# Design and Psychometric Properties of the BAtSS: A New Tool to Assess Attitudes towards Bats

**DOI:** 10.3390/ani11020244

**Published:** 2021-01-20

**Authors:** Beatriz Pérez, Boris Álvarez, Alex Boso, Fulgencio Lisón

**Affiliations:** 1Núcleo Científico Tecnológico en Ciencias Sociales y Humanidades, Universidad de La Frontera, Temuco 4811230, Chile; boris.alvarez.e11@gmail.com (B.Á.); alex.boso@ufrontera.cl (A.B.); 2Departamento de Psicología, Universidad de La Frontera, Temuco 4811230, Chile; 3Departamento de Psicología, Universidad Católica de Temuco, Temuco 4813302, Chile; 4Laboratorio de Ecología del Paisaje y Conservación, Departamento de Ciencias Forestales y Medioambiente, Facultad de Ciencias Agropecuarias y Forestales, Universidad de La Frontera, Temuco 4811230, Chile; flison@udec.cl; 5Wildlife Ecology and Conservation Lab, Departamento de Zoología, Facultad de Ciencias Naturales y Oceanográficas, Universidad de Concepción, Concepción 4070386, Chile

**Keywords:** attitudes towards bats, psychometric properties, conservation

## Abstract

**Simple Summary:**

Despite the benefits that bats offer the ecosystem, these animals are feared due to mythological beliefs and their association with dirt and disease. The COVID-19 pandemic has aggravated this situation, exacerbating the already habitual attacks on bats. Today there is an urgent need to address the human–bat conflict to develop conservation policies. Understanding peoples’ attitudes towards bats is a crucial part of this process. This study aimed to design the Bats Attitudes Standard Scale (BAtSS) and to analyze its properties. We developed a scale and analyzed its properties in a sample of 1639 Chileans. The final BAtSS consists of 34 questions on a five-point response scale. It has four factors (scientistic, positivistic, negativistic, and myths) and three subfactors (emotional negativistic, behavioral negativistic, and cognitive negativistic). The results showed that the scale was reliable and valid for measuring participants’ attitudes. Women and participants with a lower level of education were more negativistic and less positivistic. People with a higher level of education had a less mythological view of bats. We also analyzed the attitudes which would be more/less difficult to change. The BAtSS is an adequate tool and could help to understand and solve human–wildlife conflicts.

**Abstract:**

Despite the benefits that bats offer the ecosystem, these animals are feared and attacked. The COVID-19 pandemic has aggravated this situation. Today there is an urgent need to address the human-bat conflict to develop conservation policies. Understanding peoples’ attitudes towards bats are critical for this process. This study aimed to design the Bats Attitudes Standard Scale (BAtSS) and to analyze its psychometric properties. We developed an initial version of the scale in which we established the content validity; we analyzed the items and structure in a pilot sample. In the next phase, we examined psychometric properties in a sample of 1639 Chileans. The final BAtSS consists of 34 Likert-type items configured in an oblique-hierarchical structure of four factors (scientistic, positivistic, negativistic, and myths) and three facets (emotional negativistic, behavioral negativistic, and cognitive negativistic). It presents adequate internal consistency, and the analysis of concurrent validity confirms the scale’s capacity to discriminate between groups. Women and participants with a lower level of education are more negativistic and less positivistic. People with a higher level of education have a less mythological view of bats. We also analyzed the items under the assumptions of item response theory (IRT).

## 1. Introduction

In late March 2020, during the coronavirus pandemic, a group of villagers from Culden, Peru, used burning torches to attack a colony of 500 bats which lived in a cave. Alerted by the rumor that COVID-19 started when someone in China ate bat soup, the residents corralled the animals and burnt them, killing 300 specimens. The majority of the bats in the Culden colony were of the genus Myotis, insectivores which are inoffensive to human beings. Unfortunately, this is not the first time that bats have come under the spotlight when a virus affects humans. Historically, despite the many benefits they provide to the planet, bats have been subject to several stigmas, misunderstandings, and folk beliefs [1,2,3,4,5,6,7,8,9,10].

Many features make bats singular and, in some ways, surprising animals. Few people know that bats (Chiroptera) are among the most diverse orders of mammals, with more than 1300 species [11]. Some of them have developed an immune system prepared to resist different types of viruses, including coronaviruses. Understanding their biological characteristics could help the scientific community to find key mechanisms to contain COVID-19 or other viruses that can affect humans. Bats provide important ecosystem services such as pollination, seed dispersion, and pest suppression [12]. Nevertheless, society does not recognize the benefits of bats for the ecosystem [13,14,15,16]. Worse still, bats have often had to deal with the consequences of human misconceptions about them, and our irrational fears [17]. In recent decades they have become highly threatened due to anthropogenic disturbances (habitat loss, destruction of refuges, and alteration of trophic structure). The “disease avoidance” hypothesis indicates that we fear bats because they disgust us; we associate them with dirt and the propagation of disease [8,9,13,16,18,19]. This situation has been aggravated as a consequence of the COVID-19 pandemic, during which they have been a focus of media attention.

Although most studies analyze only the technical aspects of human–wildlife conflicts, peoples’ attitudes towards species can be important drivers to achieve their long-term resolution (Kingston, 2016). To understand attitudes towards endangered species is considered to be a critical step to foster their preservation [17,20]. In this regard, classical theories regarding the variables of influence on the development of human behavior, such as the theory of reasoned action and the theory of planned action [21], consider attitudes to be a fundamental element of individual decision-making.

Different studies have demonstrated the relevance of understanding how public conceptualizes species to better design animal conservation campaigns and promote public support for governmental protection policies [22,23]. Previous research has shown that this issue encompasses a huge diversity of situations and species, for example, leopard cats [24], gorillas [25], and snakes and spiders [17]. This is particularly relevant for the protection of bats [5,26,27]. As a no charismatic species, people’s perceptions and values attributed to bats are vital for their conservation [4,6,13]. The prevalence of the humanistic perspective of animals in educational campaigns [20] or the tendency to largely select flagship species based on their aesthetic appeal for conservation marketing [28,29] are two examples of a simplistic interpretation of human-wildlife relations. Only by better understanding public attitudes towards and appreciation of bats will it be possible to move conservation strategies beyond simply emphasizing affection to a broader range of human motivations for protecting bat species. Nevertheless, there is limited literature that has explored people’s attitudes towards bats, and what literature there is, suffers from a lack of consensus in the definition and operationalization of the construct.

### Measuring Attitudes towards Bats

Some works approach this construct through semi-structured interviews, open questions, or design test questions expressly for the investigation in hand with little psychometric analysis [1,3,4,8,10,18,30]. Other authors have developed specific scales, including analysis of content validity and internal consistency, mostly using Cronbach’s alpha coefficient or principal component analysis. All offer information on group comparisons based on variables such as gender, age, or level of education, reflecting the ability of the scales to discriminate between individuals [2,5,6,7,26].

Kross et al. [31] designed a scale on the perception of the following three types of animals (one per factor): bats, birds, and birds of prey. Fagan et al. [26] measured attitudes towards bats in USA, with a one-dimensional scale formed of four statements on hypothetical encounters with bats in buildings. The other quantitative assessment tools found in the literature [2,5,6,7] all follow the same line, based on the typology of the following nine basic attitudes towards wildlife and its natural habitats developed by Kellert [32,33]: naturalistic, ecologistic, humanistic, moralistic, scientistic, aesthetic, utilitarian, dominionistic, and negativistic.

Prokop et al. [6] and Prokop and Tunnicliffe [7] designed the Bat Attitude Questionnaire (BAQ) with university and child-adolescent students in Slovakia, respectively. Musila et al. [5] used an adapted version of the BAQ [6] in their study sample of Kenyans, and Barnes [2] developed the Battitude Questionnaire with students in Rodrigues Island, Mauritius, based on the information collected in different focus groups and on other scales, BAQ between them. Although these measurement instruments have a common origin, the scales resulting from their work have different structures.

The dimension negativistic is part of all scales. The scientistic and ecologistic dimensions are part of all scales, but Prokop and Tunnicliffe [7] and Barnes [2] fused these dimensions into one. Prokop et al. [6] added the dimensions of myths and knowledge about bats, although these dimensions were not part of the proposal of Kellert [34], based on their importance of scientific literature in understanding human behavior towards these animals. Nevertheless, the myths dimension did not show adequate levels of internal consistency. Prokop and Tunnicliffe [7] and Barnes [2] measured myths and knowledge about bats but similar to independent surveys. Musila et al. [5] evaluated myths as part of the scale, and knowledge similar to an independent survey. Finally, the naturalistic dimension is only present on the scale of Prokop and Tunnicliffe [7].

To summarize, despite the efforts of researchers to create an operative measurement scale with evidence of validity and reliability, a different or modified version has been used in each study, and therefore the results of each work have been based on a different conceptual model of attitudes towards bats. The consequence has been a limitation on the scope of the conclusions. We need a measurement instrument which remains constant among studies in order to obtain comparable results [35]. Rigorous analysis in the construction of the scales and at the psychometric level could help us solve this limitation [36,37,38,39,40]. Fortunately, the studies of Prokop et al. [6], Prokop and Tunnicliffe [7], Barnes [2], and Musila et al. [5] provide essential and sufficient support to guide the construction of a measurement scale that meets these types of criteria. 

In view of the above, the main objective of this study was to construct a scale of attitudes towards bats with evidence of reliability and validity in a community sample. We analyzed the factorial structure, content validity, and concurrent validity, and we estimated internal consistency. For this, we have participants from central and southern Chile, a country where bats are one of the least studied and most threatened vertebrate groups, and in which researchers demand more excellent knowledge of the species for the development of plans on the management and conservation of their populations [40]. We also identified which attitudes are easiest and most difficult to modify for each dimension. In addition, we reported the Chilean attitudes towards bats by sex and level of education and advocate for using, in the academic community, the same conceptual model and instrument to measure attitude towards bats as a base for enhancing conservation efforts.

## 2. Materials and Methods

The construct, i.e., attitudes towards bats, is defined as relatively stable evaluations of bats, either positive or negative, at the cognitive, affective, or behavioral level [41]. In view of their importance in the literature, we selected four of Kellert’s nine attitude types [33,34] for the theoretical definition of the dimensions of the assessment instrument as follows: (1) Scientistic, “primary interest in the physical attributes and biological functioning” of bats [33] (p. 179); (2) ecologistic, primary concern for bats and their natural habitats [33]; (3) utilitarian, “primary concern for the practical and material value” of bats and their animal habitat [33] (p. 179); and (4) negativistic, “primary orientation an active avoidance of (bats) due to dislike or fear” [33] (p. 180). We also included a fifth component, i.e., myths, defined as people’s beliefs, legends, or non-scientific knowledge about bats, based on the interest shown by researchers in the literature [5,6,7].

Following the recommendations for the cultural adaptation and validation of questionnaires, and for test construction [37,39], we carried out a retro-translation process of the items of the existing questionnaires [2,5,6,7,31] and we formulated other new items using the most appropriate vocabulary within the context of the Chilean culture. Some items required modification as part of the process of linguistic and cultural adaptation. We finally obtained an initial version of 53 items (Table A1), i.e., 12 for the scientistic dimension, 18 for the negativistic dimension, 12 for the ecologistic dimension, 5 for the utilitarian dimension, and 6 more for the dimension of myths.

The measurement instrument was subjected to the judgement of experts to obtain evidence of content validity. A table containing the definitions of the dimensions and the list of the 53 items, categorized according to the theoretical dimension of appropriateness, was given to 10 judges who were experts in the areas of ecology and conservation, sociology, psychology, and attitude measurement. The judges had to answer the following questions: “Do you believe that this item is suitable for measuring the dimension indicated?” and “Do you believe that any important item or concept is missing to represent this dimension completely?” In the case of disagreement, they were asked to make a recommendation. The experts also evaluated the wording of the instructions and the items. After analysis of the answers, and obtaining agreement between the judges by group discussion, the wording of some items was modified and two of the negativistic dimensions were eliminated as redundant (see Table A1).

A scale of 51 items (Table A1) was subjected to a pilot study with a sample of 67 university students. The researchers went to the classrooms to ask students to participate. Those who agreed to participate on a voluntary, confidential basis replied in situ to the questionnaire, which had been distributed previously online. The university students evaluated the online format and the wording of the instructions and the items. Subsequently, they commented to the researchers on the difficulties which arose. In conjunction with theoretical criteria, the statistical analysis of the items in this pilot study guided us in selecting each dimension’s best items. This selection made up the preliminary scale of 39 items.

Several items were discarded (see Table A1) to present a lower discriminatory capacity and contribution to the scale’s consistency than the rest of the items of its dimensions (five in the scientific dimension, three in the negativistic dimension, and four in the ecological dimension). In the case of the myths dimension, five of the six items had a deficient discriminatory capacity (corrected item-total correlation less than 0.3) and low contribution to the consistency of the scale, i.e., the Cronbach’s alpha improves if the item is deleted. Nevertheless, considering the theoretical interest of this dimension and the fact that the low internal consistency may have been influenced by the small sample size, the six items of this dimension were kept for the following phase of this study.

Thirty-nine of the 51 items were retained. These were subjected to exploratory factor analysis (EFA) with varimax rotation and principal component extraction. The resulting 6 dimension factorial structure did not agree exactly with the theoretical structure as follows: (1) the scientistic dimension was identical to the theorized dimension; (2) a second dimension emerged in the EFA which we called positivistic, combining the items that correspond to the theoretical dimensions ecologistic and utilitarian; the negativistic theoretical dimension was divided into three, according to the attitude definition structure, and were (3) emotional negativistic, (4) behavioral negativistic, and (5) cognitive negativistic; and (6) the dimension of myths was maintained. Finally, a preliminary scale of 39 items passed to the next phase of the work.

### 2.1. Participants

A sample of 2189 participants was collected by non-probabilistic convenience sampling. After eliminating the incomplete questionnaires, questionnaires with response time suspected of fraud, the outlier cases, and those which did not meet the selection criteria, the final sample of 1639 participants was obtained consisting of all Chilean nationals resident in the country, aged over 18 years and not related with professional or productive agriculture. Table 1 shows the descriptive data.

### 2.2. Measurements

The sociodemographic questionnaire includes sociodemographic questions for sample characterization, such as age, sex, educational level, area of study, nationality, and religion.

The Bats Attitudes Standard Scale (BAtSS), preliminary version consists of a scale of 39 items (Table A1) with the following five response options: (1) totally disagree, (2) disagree, (3) neither agree nor disagree, (4) agree, and (5) totally agree. We proposed the theoretical structure of six dimensions defined above as follows: scientistic (7 items); positivistic (11 items, a combination of dimensions theorized as biological and utilitarian, primary concern for bats, their natural habitats and, their practical and material value); emotional negativistic (4 items); behavioral negativistic (4 items); cognitive negativistic (7 items); and myths (6 items).

### 2.3. Procedure

We used the internet-mediated research (IMR) technique, which focused on the role of the internet as a means of conducting research [42]. The sample was collected online from August to November 2019, using an electronic survey tool, SurveyMonkey (www.surveymonkey.com). There was substantial dissemination, due to the link to the survey being broadly shared on social media via Twitter, Facebook, Instagram, Linkedin, Whatsapp, email, and through other channels. The link was shared from the project’s accounts on these social network sites and channels with national organizations related to the environment and conservation, as well as researchers from other universities in the country, who in turn shared this link. All the participants gave their voluntary consent to participate in the study. The study’s participants were anonymous, and the researchers never collected the names of the participants or contacted them directly. The project did not require approval by an Ethics Committee. The time taken to complete the questionnaire and assessment instrument was approximately 15 min.

### 2.4. Data Analysis

The correlational methods used to determine the psychometric characteristics of a measurement instrument are very sensitive to study samples; that is, sampling errors and other random variations may affect the results. Consequently, it is advisable to carry out a cross-validation or replication process of the factors in new samples to establish their generalizability [38]. Due to the above, the sample was randomly divided into two parts to identify the best items and the factors of the measurement instrument in the first half, and to be able to replicate these factors in the second half; the first group consisted of 820 participants, and the second group consisted of 819 participants.

The first sample was used to analyze the discriminative capacity of the items by the corrected item-total correlation, the normality of the scores by the Kolmogorov–Smirnov (*K-S*) test, and the levels of kurtosis and asymmetry. This procedure was used to identify and discard the items on the scale that did not differentiate the participants based on their attitudes towards bats. We also explored the inter-item correlation per dimension by Pearson’s correlation, to detect extreme levels of correlation. Correlations that are too high (over 0.8) are an indicator that the items are too similar, and therefore measure the same behavior. One of them should be removed due to redundancy. Correlations that are too low (below 0.3) are an indicator that one item (or several) is not measuring the same construct as the rest. This item should be removed. Finally, once the relevance of the data for factorial analysis was established by Bartlett’s index and the Kaiser–Meyer–Olkin (KMO) test, this first sample was subjected to EFA using the unweighted least squares extraction method and oblique oblimin rotation.

The six-factor structure, obtained in the first group, was contrasted with the second group. The analytic strategy used was confirmatory factor analysis (CFA), which considered the robust unweighted least squares (ULSMV) estimator on a polychoric matrix, due to the ordinal nature of the data. The CFA allowed us to confirm that the result obtained through the EFA (the existence of six different factors in this measurement instrument) was replicated or adjusted in another study sample (in this case, in the second half of the sample). In addition, in order to determine whether this factorial structure corresponded to a correlational or hierarchical model, we explored the fit of an oblique six-factor model (M1) and a second-order hierarchical model of six factors (M2) [43]. In response to the theoretical relation among the emotional negativistic, behavioral negativistic, and cognitive negativistic factors, we explored an oblique solution which we called the hierarchical-oblique model (M3), and a hierarchical solution which we called the third-order hierarchical model (M4). In both M3 and M4 we included a fourth factor that we called negativistic, consisting of three facets, i.e., emotional negativistic, behavioral negativistic, and cognitive negativistic. In M3, this factor correlated with the scientistic, positivistic, and myths factors, while in M4 the four factors presented a higher order factor (see Figure A1).

Finally, we included analysis of the structure of this assessment tool as a bifactor model (see Figure A1), based on the model with best fit to the four options named above (M5) [40,43]. This model explores the coexistence of a general factor that explains common covariance between all the variables observed, and the factors made up of the items with a higher shared variance. Fitting a bifactor model enabled us to determine the existence of a sufficiently strong general factor to justify a global score, as well as scores for each individual factor [40,44]. The comparison of the goodness of fit indices for each model allowed us to evaluate which model best fits the data and select the better.

The goodness of fit indices based on the statistical significance of *X^2^* to answer the question, to what extent is each of the models considered to be adequate. The adjustment indices considered were the root mean square error of approximation (RMSEA), which is a measurement based on approximation errors which assesses whether the model fits “roughly” well in the population and the comparative fit index (CFI) and the Tucker–Lewis index (TLI) comparative fit indices based on the comparison of the approximation error of the proposed model and the independence model. A CFI and TLI ≥ 0.95 and RMSEA < 0.05 were considered to be a good fit; a CFI and TLI ≥ 0.90 and a RMSEA < 0.08 were considered to be acceptable [45]. Furthermore, to assess to what degree the data could be considered to be an essentially one-dimensional structure in M5, the omega hierarchical coefficient (OmegaH), the percent of uncontaminated correlations (PUC), and the explained common variance (ECV) were used. OmegaH values above 0.80 are recommended for the total scores to be considered essentially one-dimensional [44]. ECV and PUC values higher than 0.70 indicate a slight relative bias, when the scale is considered to be essentially one-dimensional [40].

Below, we report the correlation among the factors in the scale to check that the factors are related expectedly and rule out multicollinearity, that is, that some factors measure the same construct when, being different factors, they should measure different constructs, although related. Suspicions of multicollinearity were discarded with tolerance values above 0.1 and variance inflation factor (VIF) values below 10. The reliability of the scale, in both the first and second sample, was calculated by ordinal alpha and McDonald’s omega, internal consistency coefficients which are best suited to ordinal data. The fitted values must be higher than 0.70 [40].

Finally, with a random sample of 1000 participants, we analyzed the data under the assumptions of item response theory (IRT) and took the Samejima graduated response model [46]. This is a particular case of the two-parameter logistic model. It indicates the capacity of each item to discriminate the estimated score of each participant in the factor (parameter a). The higher the score in this parameter, the greater the discriminative capacity. It also indicates the probability of selecting a specific response category or a higher category for a given level of the measurement variable, i.e., for each response option of each item (parameter b). According to the cumulative process defined by [46], the difference b4 − b1 is indicative of the difficulty/ease of changing from one extreme to the other in the measurement levels, i.e., to change from totally disagree to totally agree or vice versa. The higher the score, the greater the difficulty to modify the attitude represented by the item in the participants.

To analyze the concurrent validity, we used the complete database (N = 1639). The K–S test was used to show that there was no normal distribution of the univariate data for the total score in the scale factors. Levene’s test showed that a few cases did not comply with the assumption of homoscedasticity. We selected Welch’s t-test or unequal variances t-test to compare the means of two groups of different sizes. As a statistical technique for comparing more than two groups, and due to the non-fulfillment of the parametric assumption of homogeneity of variances, we used the Welch’s F test, robust test to the non-fulfillment of this assumption, in conjunction with the Games–Howell post hoc test. In both cases, we used Hedges’ g (g) to calculate the size of the correction effect for groups of different sizes. Values up to 0.2 report a small effect, from 0.2 to 0.4 an intermediate effect, and 0.4 or higher a large effect 39. The software used for this study was SPSS 24.0, FACTOR 10.9, Excel, Mplus 7.11, R, and IRTPRO. To calculate the bifactor indices, we used the Bifactor Indices Calculator [47].

## 3. Results

### 3.1. Preliminary Analyses

None of the items have a normal distribution. Of the 39 items, five presented an asymmetry index greater than two, and of these five, two had a kurtosis index higher than seven (Table A3). Nonetheless, the items considered to be uninformative because they did not discriminate between subjects were those with a frequency of 95% or higher in one response option. The five items with high asymmetry or kurtosis (Items 23, 24, 27, 31, and 32) presented a maximum response frequency between 75.1 and 86.7% in one of the five possible answer options. This is indicative of low discriminative power, but not nil. Therefore, it was decided to retain them as part of the measurement instrument.

The item-total correlation per factor produced a value of 0.27 for Item 19, corresponding to the cognitive negativistic factor, and therefore it was eliminated from the questionnaire. The rest of the items presented an item-total correlation per factor higher than 0.38 (corrected item-total correlation less than 0.3 is indicative of insufficient discriminatory capacity).

Two items of the myths dimension had low values, correlating with the remaining items in the dimension (between 0.14 and 0.30). The correlation between these was suitable (0.52). This is because these two items measure positive myths while the rest measure negative myths, causing confusion in the dimension. Those who disagree with a mythological view of bats score low in all the items, but people who agree with this view do not give a uniform response. People who agree with positive myths tend to disagree with negative myths and vice versa. For this reason, we decided to eliminate Items 22 and 25.

Finally, we studied the high correlations to detect possible multicollinearity and identify redundant items (items representing the same idea). We eliminated two items belonging to the positivistic factor because they presented correlations higher than 0.8 with other items, i.e., Item 8, which presented a correlation of 0.87 with Item 9 and Item 10 with a correlation of 0.82 with Item 9 and 0.83 with Item 11. Table A3 presents descriptions of the 34 items that were finally included in the scale.

### 3.2. Exploratory Factor Analysis

The relation between the variables is high, therefore, the quality of the data for applying EFA is good (KMO = 0.943). Bartlett’s test indicates that the correlation matrix differs from the identity matrix, χ^2^ (561) = 16,874.928 with a significance level *p* < 0.001. These values are indicative that the study data has sufficient quality to apply EFA. Consequently, we did an EFA with the 34 selected items. As a result, we obtained a structure of six factors which explained 57.6% of the variance. This structure is consistent with the structure obtained in the pilot sample, except for Item 39. This item is part of the cognitive negativistic dimension, instead of the emotional negativistic dimension. Table 2 presents the factorial weights of the items in each of the dimensions.

### 3.3. Confirmatory Factor Analysis

Here, we try to corroborate the fit of the six-factor model resulting from the EFA by CFA with the Group 2 sample (N = 819). The Syntaxis Mplus for the factorial models subjected to CFA are listed in Table A2.

All the models present an adequate fit (Table 3), except the bifactor model which presents an optimum fit. Nevertheless, the OmegaH value is 0.24, and therefore the total score cannot be considered to be essentially one-dimensional. In addition, although PUC produces a value of 0.834, the ECV is equal to 0.14. These values indicate that considering the assessment instrument as essentially one-dimensional involves a very high relative bias. The bifactor model was discarded, and with it the possibility of considering a global score for the scale. The oblique (M1) and hierarchical-oblique (M3) models were preferred for their better fit as compared with the second- (M2) and third- (M4) order hierarchical models. M1 and M3 have lower values of RMSEA and higher values of CFI and TLI than M2 and M4. The M1 and M3 have a nearly identical fit. Although the oblique model (M1) is more parsimonious, we preferred the hierarchical-oblique model (M3) for its greater theoretical coherence. The instruments presented in the literature refer to a single negative factor, which is not represented in M1. Figure 1 presents the factorial weights of the items in the different factors and facets. This factorial structure implies that we can independently consider the scores in the four factors (scientistic, positivistic, negativistic, and myths) and in the three facets (emotional negativistic, behavioral negativistic, and cognitive negativistic).

The correlations between the factors are significant and in the expected direction. The following correlations are negative: scientistic with negativistic (−0.461) and myths (−0.280); positivistic with negativistic (0.816) and with myths (−0.500), except between the pairs scientistic and positivistic (0.573), and negativistic and myths (0.765). The correlation between the negativistic and positivistic factors is higher than 0.8, which raises a suspicion of multicollinearity. Nevertheless, the level of tolerance between the two scores is 0.546, and the VIF is 1.83, therefore, we can discard this suspicion. These results indicate that the factors of the instrument measure different but correlated constructs.

### 3.4. Internal Consistency

McDonald’s omega and the ordinal alpha coefficients indicate that the assessment tool presents adequate internal consistency for the four factors and the three facets (see Table 4). This result is indicative that it is possible to infer the real score of the participants in the theoretical constructs through the score obtained in the factors of this measurement instrument.

### 3.5. Discriminative Capacity and Difficulty of the Items

Table A4 presents the results of the Samejima graduated response model. The higher the score in b4 − b1, the greater the difficulty to modify the attitude represented by the item. Considering the factors and facets representing negative evaluations of bats, we find that the items that measure the most challenging participants’ attitudes to modify are the following: Item 29 in emotional negativistic; Items 34 and 36 in behavioral negativistic; and Items 20, 26, and 33 in cognitive negativistic. In the myths dimension, the items that represent the ideas (attitudes) that are most difficult to modify in the sample individuals are Items 21 and 23.

### 3.6. Concurrent Validity: Differences between Groups according to Gender and Educational Level

In relation to the mean score and score interval for each factor and facet, we observed that the total sample shows high scientistic (scoring interval (SI) = 0–35, mean (M) = 25.05, and standard deviation (SD) = 5.74) and positivistic attitudes (SI = 0–45, M = 35.17, SD = 5.58), as well as low negative (emotional negativistic SI = 0–15, M = 7.30, and SD = 3.03; behavioral negativistic, SI = 0–20, M = 5.78, and SD = 2.64; cognitive negativistic, SI = 0–35, M = 16.33; and SD = 4.88) and mythical attitudes (SI = 0–20, M = 5.77, and SD = 2.46). However, we observed that some participants differ from others in their attitude levels towards bats, according to gender and educational level. Men present a higher mean score in the positivistic factor (*g* = −0.36) and women in the negativistic facets (emotional negativistic, *g* = 0.33; behavior negativistic, *g* = 0.24; and cognitive negativistic, *g* = 0.39). In both, the size of the effect is small. Statistically significant differences are found for the scientistic and myths factors, but the Hedges’ g statistic indicates that these have a zero effect (see Figure 2).

Statistically significant differences are found in level of education (see Figure 3) with a small size of effect in the positivistic dimension (*g* = 0.27) and an intermediate effect in the scientistic dimension (*g* = 0.41) and the cognitive negativistic facet (*g* = 0.41). The other significant differences present a nil size of effect. In the scientistic dimension, the participants with postgraduate education (high level) present a lower mean score than participants with university or professional/technical education (*p* < 0.001, medium level), and with primary or secondary education (*p* < 0.001, low level). The latter present lower mean scores in the positivistic dimension than the participants with a medium level education (*p* = 0.046) and the participants with a high level of education (*p* = 0.001). Finally, the participants with a high level of education have a lower mean score in the cognitive negativistic facet than the participants with a medium level of education (*p* = 0.005) and the participants with a low level of education (*p* < 0.001); while participants with a medium level of education have a lower mean score than those with a low level of education (*p* = 0.016).

## 4. Discussion

Considering the importance of attitudes in the development of conservation behaviors [17,20], especially in the case of animals which cause fear and evoke disgust; are associated with dirt, disease, or infection; or are aesthetically less favored [4,6,7,13,48,49], and in the absence of a conceptual model and a unified measurement instrument, we developed the BatSS in a Chilean community sample to evaluate attitudes towards bats. For this, we counted on the contributions of previous measurement systems [2,5,6,7]. After studying the content validity and initial analysis in a pilot sample, we analyzed items and the factorial structure of the scale using a crossed validity strategy. We examined the internal consistency and concurrent validity. With the latter, we also examined attitudes towards bats according to gender and educational level. Furthermore, we identified the attitudes which were the most difficult to change in the study sample.

After analysis of the items, we obtained a scale of 34 Likert-type reagents with discriminative capacity, with an oblique-hierarchical factorial structure and good values of internal consistency. The BatSS presents four correlated factors. Two of them represent a positive evaluation of bats (scientistic and positivistic), and the other two a negative evaluation (negativistic and myths). The negativistic dimension is hierarchical, as it is subdivided into three facets (emotional negativistic, behavioral negativistic, and cognitive negativistic). This theoretical model differs from others in the literature, apart from the scientistic dimension, which maintains high coherence levels with earlier works [2,5,6,7].

The positivistic dimension is composed of items referring to concern for conservation of bat species from an ecologistic perspective; these are common items in other scales under the headings of the ecologistic dimension [5,6] or the eco-scientistic dimension [2,7]. It also considers positive evaluation of bat species, based on their importance in ecosystem functioning and farming. This is an innovative contribution based on the utilitarian typology of Kellert [33,34], which makes it possible to capture results obtained in other works investigating the relation between positive attitudes and recognition of the benefits of bats for the ecosystem [4]. Nevertheless, the IRT analyses indicate that the items on ecologistic attitudes have a greater capacity to discriminate the score of each participant in the factor and that the utilitarian ideas covered by the rest of the items are more difficult to change. For example, Item 14, “bat excrement is a source of good fertilizer for farming” or Item 18, “the activity of bats gives added value to crops in the market”. Therefore, promoting knowledge about bats that contribution to our wellbeing is identified as an essential focus to encourage change in attitudes towards bats.

The negativistic dimension is present in the theoretical models of all the assessment tools reported in the literature [2,5,6,7]; in the BatSS, however, this dimension has a hierarchical structure which enriches the measurement. Individuals are classified differentially into emotional, behavioral, and cognitive negative attitudes, allowing different negativistic profiles to be identified based on the combinations of the scores in the three subscales. In addition, if we carefully review the items’ wording, we observe a trend in the IRT results’ analyses, i.e., the most challenging ideas to change are those that identify a more immediate danger to humans. For example, Item 29, “bats are dangerous for humans”, Item 20, “bats’ activity contaminates crops”, or Item 33, “bats can be dangerous for domestic animals”. Consequently, we identify the deconstruction of the collective imaginary about the bat as a threat to humans as a focus of relevant conservation intervention. This result is consistent with the “disease avoidance” hypothesis [8,9,13,16,18,19]. Fortunately, the most extreme negative behaviors are the easiest to change, such as Item 31, “bats should be exterminated” or Item 32, “we should attack bats”.

Finally, researchers have had varying successes in capturing attitudes to mythology. Despite the difficulties in incorporating this dimension into the measurement instruments, the investigators have a persistent interest in the mythological view due to the complexity of modifying behaviors based on mythological beliefs [2,7]. Myths and folk stories have led to people lighting fires in bats’ caves, killing them, and even capturing them to use in supposed cures for diseases [8]. This dimension was incorporated into the BAQ of Prokop et al. [6], although it did not achieve adequate levels of internal consistency. Musila et al.’s version of the same measurement instrument [5] obtained acceptable levels in this dimension, while Barnes [2] and Prokop and Tunnicliffe [7] evaluated myths about bats as an independent measurement, not as part of the measurement of attitudes. Fortunately, in the BatSS, we obtained a myths dimension with a good internal consistency level [38]. Nevertheless, its cultural nature complicates extrapolation to other contexts, and therefore special attention would be needed for use with other populations.

Our results indicate that, in general, Chileans do not have extremely negative attitudes towards bats. Despite this, the scientific literature demands to promote more excellent knowledge and acceptance of the species in Chilean populations [40]. We found differences in attitudes by sex and level of education, so the scale has a discriminative capacity between groups, and therefore concurrent validity [37,38,39]. These results agree in part with the scientific evidence, although this comparison is subject to limitations due to the disparity of theoretical models, measurements and cultural contexts between studies. Musila et al. [5], Prokop and Tunnicliffe [7] and Prokop et al. [6] found that women had higher scores in negative and mythological attitudes [5,6,7], although, in most cases, it was men who carry out aggressive actions against bats [5]. Barnes [2] found no differences by gender in his study sample. The literature indicates that a greater knowledge of biology or bats and a higher level of education support a more positive and less mythological evaluation [2,4,5,6,7,10].

In our study sample, the myths dimension showed no difference by sex and education level, which may indicate that the mythological view is less firmly rooted in the sample. Although the scientific literature refers to Chile’s mythological beliefs, these are ascribed to specific ethnic groups or subcultures [50,51]. However, women have a more negative attitude than men on emotional, cognitive, and behavioral levels. The social norms of gender support these results, i.e., women are oriented towards the private sphere and a more significant concern for care and nutrition. Therefore, a more threatening vision by women is expected [52]. In the case of level of education, the only negative attitude that established differences was cognitive, i.e., the lower the level of education, the stronger was the negative cognitive attitude. This is an interesting finding, since it is more difficult to change emotional and behavioral attitudes [41]. Finally, we observed a similar tendency in the rest of the results, except that scientific interest in bats was lower among participants with postgraduate studies. We interpret this finding to be the result of more strongly developed and defined scientific interests in individuals of this group whose studies were unrelated to bats.

Increased scientific interest and primary concern for natural habitats and the animals’ practical and material values increase conservation behaviors [53]. The same is true when negative and mythological beliefs are reduced [25]. Consequently, studies have indicated that considering local people’s attitudes toward wildlife was necessary for effective conservation [49,54]. Studies have also shown that national conservation attitudes reflected wildlife management and policy from countries such as the USA or Australia [22]. Definitely, the BAtSS scale emerges as a useful tool to know individuals’ attitudes in the community, examine changes in their attitudes, and differentiate social groups based on their attitudes. This information can be input for detecting groups and individuals that pose an imminent danger and for designing better animal conservation campaigns at the community level that encourage changes in attitudes [23], and therefore conservation behaviors [21].

We identified the lack of representativeness of the sample as a limitation of this work. Furthermore, the cultural nature of the myths dimension would hinder the adaptation and validation of the BatSS in other cultural contexts than Chile. We also think that we need to continue accumulating evidence of validity in Chile to increase the solidity of the scale. For example, in future work, the BatSS scale should be studied to definitively confirm its internal structure, as well as to analyze the convergent validity of the scale based on other theoretical constructs, for example, knowledge about bats.

## 5. Conclusions

The current situation in which bats are persecuted generally around the world requires an effort by investigators to address the human–bat conflict to favor the development of conservation policies. People’s attitudes towards these animals have proven to be of key importance in this process. We identified the need to construct a tool that investigators could use to measure this theoretical construct in a specific cultural context, but which, potentially, would be adaptable to other sociocultural scenarios [38].

In this study, we designed the BatSS. This scale, and the theoretical model on which it is based, provide a tool with proven validity and reliability for measuring attitudes towards bats in the Chilean community population. Its adequate psychometric properties make it a good candidate for adaptation to other cultural contexts. Extension of its use would facilitate replication, which is the scientific gold standard for the confirmation of results [55]. This framework extends practical understandings of conservation conflict interventions by offering a novel, transdisciplinary, diagnostic tool for better understanding their complex, multifaceted variables [56].

## Figures and Tables

**Figure 1 animals-11-00244-f001:**
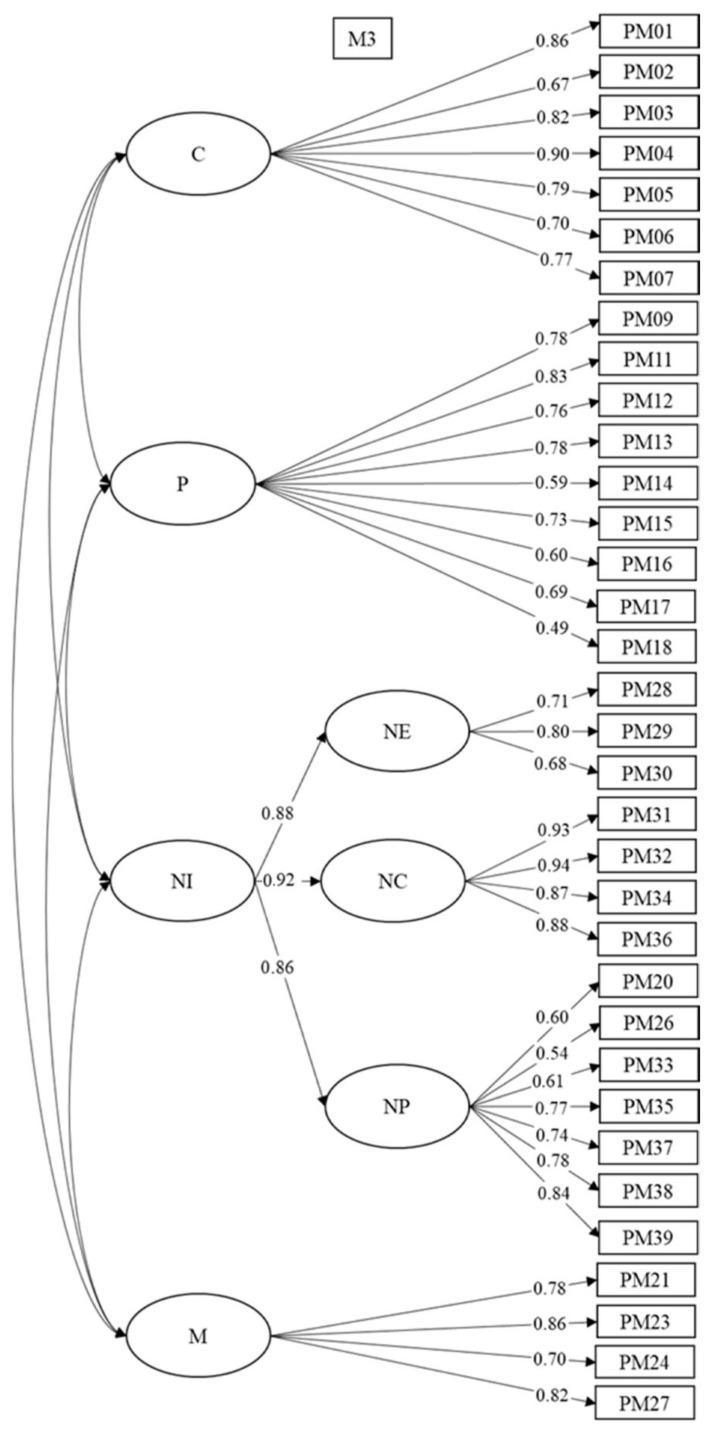
Factorial weights of the items in the hierarchical-oblique model (M3). S, scientistic; P, positivistic; N, negativistic; EN, emotional negativistic; BN, behavioral negativistic; CN, cognitive negativistic; M, myths.

**Figure 2 animals-11-00244-f002:**
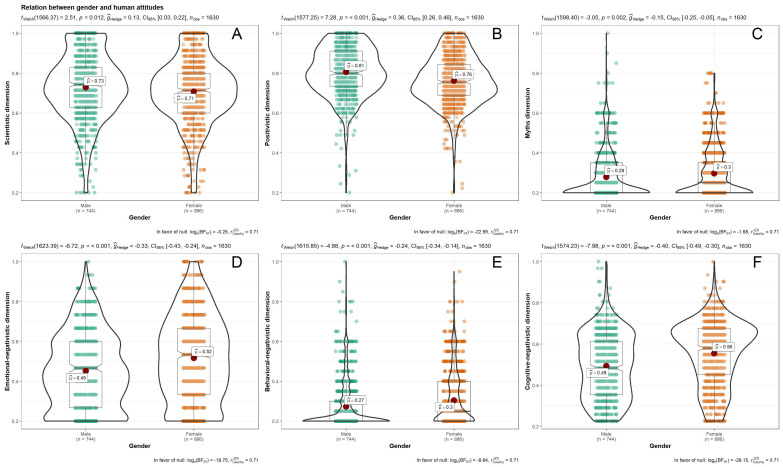
Difference of means between men and women in the factors of the Bats Attitudes Standard Scale (BAtSS): (**A**) = Scientific dimension; (**B**) = Positivistic dimension; (**C**) = Myths dimension; (**D**) = Emotional negativistic dimension; (**E**) = Behavioral negativistic dimension; (**F**) = Cognitive negativistic dimension.

**Figure 3 animals-11-00244-f003:**
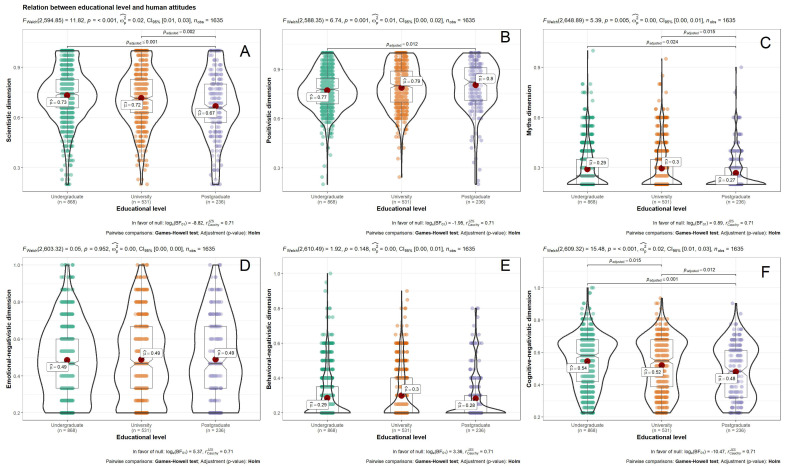
Differences of means in the factors and facets of the BAtSS, according to educational level: (**A**) = Scientific dimension; (**B**) = Positivistic dimension; (**C**) = Myths dimension; (**D**) = Emotional negativistic dimension; (**E**) = Behavioral negativistic dimension; (**F**) = Cognitive negativistic dimension.

**Table 1 animals-11-00244-t001:** Sociodemographic characteristics of the participants.

Variables	n	%
Gender (N = 1635)		
Male	744	45.5
Female	886	54.2
Other	5	0.3
Age (N = 1642)		
≤29	1194	72.8
30–59	405	24.5
≥60	43	2.6
Geographical areas (N = 1.631)		
Center	1473	90.3
South	158	9.7
Level of studies (N = 1635)		
Secondary education or less	868	53.1
University or professional technician	531	32.5
Postgraduate	236	14.9
Religion (N = 1638)		
Christians	685	41.8
Other religions	89	5.4
Atheist	234	14.3
None	630	38.4
Have you seen a bat in person? (N = 1639)		
Yes	1141	69.9
No	498	30.4

**Table 2 animals-11-00244-t002:** The exploratory factor analysis (EFA) results in Group 1 sample (N = 820). Extraction of unweighted least squares and oblimin oblique rotation.

Items		Factors					
		1	2	3	4	5	6
Scientistic							
1	I would like to learn more about bats	**0.650**	−0.030	−0.039	0.027	−0.051	−0.145
2	Knowing about the activity of bats is important for me	**0.739**	−0.072	0.075	−0.121	−0.034	−0.055
3	I would like to take part in a trip or a congress, or other activity, to learn about bats	**0.820**	−0.001	−0.038	−0.021	0.009	0.107
4	It would be interesting to take part in a scientific activity about bats	**0.818**	0.020	−0.044	0.045	0.034	0.004
5	I would like to exchange knowledge about bats with other people	**0.875**	0.014	−0.018	−0.017	−0.009	0.086
6	It would be interesting to be able to teach others about bats	**0.829**	0.004	0.019	−0.010	−0.010	0.083
7	I would like to read a scientific article or see a documentary about bats	**0.701**	−0.003	−0.023	0.065	0.019	−0.099
Positivistic							
9	Bats are important for the functioning of our ecosystem	0.055	**−0.735**	0.066	0.133	0.089	−0.154
11	Humans should protect bats	0.018	**−0.712**	−0.073	0.268	0.050	0.003
12	Spaces should be set aside for bat conservation in farmland	0.084	**−0.702**	−0.047	0.084	−0.029	0.067
13	Humans must learn to coexist with bats	0.015	**−0.701**	−0.106	0.181	0.054	−0.006
14	Bat excrement is a source of good fertiliser for farming	0.037	**−0.528**	−0.127	−0.085	−0.022	0.031
15	Bats help in the biological control of pests	0.013	**−0.726**	0.095	0.082	−0.027	−0.131
16	Bats help food security	0.001	**−0.703**	0.014	−0.184	−0.154	0.034
17	Some species of bat help to disperse tree seeds	0.062	**−0.600**	−0.021	−0.032	−0.056	−0.050
18	The activity of bats gives added value to crops in the market	0.030	**−0.645**	−0.006	−0.206	−0.138	0.055
Emotional Negativistic							
28	Bats are ugly	−0.120	0.021	**0.708**	−0.022	−0.103	0.046
29	Bats are dangerous for humans	0.044	0.172	**0.564**	−0.032	0.180	−0.023
30	I am afraid of bats	−0.064	−0.007	**0.644**	−0.036	0.053	0.076
Behavioral Negativistic							
31	Bats should be exterminated	0.013	0.205	0.138	**−0.572**	0.111	0.142
32	We should attack bats	−0.021	0.092	0.078	**−0.526**	0.138	0.307
34	Bat refuges should be eliminated to prevent them from breeding (block up caves, cut down trees, etc.)	−0.067	0.031	0.153	**−0.419**	0.357	0.100
36	We should stop bats from reproducing	−0.044	0.122	0.026	**−0.478**	0.464	0.067
Cognitive Negativistic							
20	Bats’ activity contaminates crops	−0.045	0.164	0.092	0.142	**0.386**	0.073
26	Bats attract other species of rodents	0.006	−0.008	0.024	0.056	**0.405**	0.207
33	Bats can be dangerous for domestic animals	0.026	0.002	0.353	0.023	**0.385**	−0.079
35	Bats contaminate water resources	−0.075	0.030	0.114	−0.057	**0.669**	−0.014
37	Bats damage machinery/buildings	−0.060	0.037	−0.016	−0.112	**0.741**	0.034
38	Bats harm agriculture	−0.040	0.218	−0.109	−0.094	**0.749**	−0.042
39	Bats are aggressive	−0.019	0.051	0.250	−0.102	**0.507**	0.016
Myths							
21	The bat is a symbol of ill omen	0.053	0.105	0.212	0.030	0.086	**0.373**
23	When you see a bat, it is a sign that someone wants to harm you	0.029	0.002	0.086	0.113	0.051	**0.820**
24	Bats become vampires	0.004	0.046	−0.062	−0.069	−0.046	**0.727**
27	Bats should be burnt to prevent witchcraft	−0.093	−0.040	−0.013	−0.180	0.035	**0.738**

The highest factorial weight for each item among the 6 factors is marked in bold.

**Table 3 animals-11-00244-t003:** Fit indices of models subjected to confirmatory factor analysis (CFA).

Models	χ^2^	df	CFI	RMSEA	TLI
Oblique (M1)	160.3	512	0.939	0.051 (0.048 0.054)	0.933
Second-order hierarchical (M2)	1917.17	521	0.922	0.057 (0.054 0.060)	0.916
Hierarchical-oblique (M3)	1642.3	518	0.937	0.051 (0.049 0.054)	0.932
Third order hierarchical (M4)	1927	520	0.921	0.057 (0.055 0.060)	0.915
Bifactor (M5)	1177.7	484	0.961	0.042 (0.039 0.045)	0.955

M1, Model 1; M2, Model 2; M3, Model 3; M4, Model 4; M5, Model 5; χ^2^, Chi-square; df, degrees of freedom; CFI, comparative fit index; RMSEA, root mean square error of approximation; TLI, Tucker–Lewis index.

**Table 4 animals-11-00244-t004:** Coefficients of McDonald’s omega and ordinal alpha for factors and facets.

		S	P	N	EN	BN	CN	M
Omega McDonald´s	Group 1 (N = 820)	0.938	0.936	0.940	0.812	0.943	0.888	0.907
	Group 2 (N = 819)	0.920	0.895	0.932	0.784	0.947	0.871	0.876
Ordinal Alpha	Group 1 (N = 820)	0.937	0.936	0.939	0.810	0.942	0.882	0.900
	Group 2 (N = 819)	0.919	0.893	0.950	0.775	0.945	0.867	0.869

S, scientistic; P, positivistic; N, negativistic; EN, emotional negativistic; BN, behavioral negativistic; CN, cognitive negativistic; M, myths.

## Data Availability

The data presented in this study are available on request from the corresponding author. The data are not publicly available due to restrictions on privacy.

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
