# Peer review of "Design and Psychometric Properties of the BAtSS: A New Tool to Assess Attitudes towards Bats"

_animals, 2021, doi:10.3390/ani11020244_

Round 1

Reviewer 1 Report

Summary

The authors seek to create a tool to evaluate attitudes towards bats in order to offer a standardized method for bat conservation. There is so much focus on the creation process of the tool, yet the manner in which it is written makes it difficult to evaluate. There is a lack of elaboration on the specific ways in which it can impact and improve on not just bat conservation, but conservation more broadly, resulting on a rather limited audience who would be interested in this paper.

While this is a high need area, the presentation of how the final items were included in the tool had many steps unexplained in the Methods, and it is difficult to evaluate whether the statements and models included were vetted properly and apt to reach those selections. There is not proper list of all the items considered anywhere in the manuscript. There is also much about the population chosen that is not described and may lead to the need to explain gaps in the Discussion. The Results are presented in a way that repeats many of the statements of the Tables, and there are far too many Tables in the main text for things that would better fit in Supplement Materials. There should have been more in the Discussion about the manner in which the results of each of the six dimensions can have on conservation efforts and what kind of actions may best serve biologists based on these results. The impact of having such a tool and how it may affect the issues that challenge species persistence of bats is not discussed in great detail nor how it may be important to conservation more broadly.

Introduction

Line 58 – There are more than 1400 species of bats, and the citation chosen is inappropriate for citing even the 1300 species the authors list. The most up-to-date vetted taxonomic database of described bat species is on batnames.org, and there is a citation format for the website. It is a more updated version of the previously oft cited Simmons (2005) for total number of bat species.

Line 61 – I think the use of the word “control” regarding COVID-19 may not be quite accurate in relation to what studying bat biology may help us with. I think “contain” or something less directed than “control” is more accurate.

Line 62 – Citation needed for bats and their ecosystem services.

Line 63 – “Almost all species of bats therefore enjoy high levels of institutional protection” –That statement is not true. There are very few bat species that are protected by national laws in various countries, and certainly few on the international level via multilateral agreements.

Lines 82-116 – These paragraphs give so much specific unnecessary details on these two other studies. I think the main point here could be more succinct—it seems that it was supposed to point out that despite using similar starting points, the details in the surveys were broken down in different ways from one another, leading each to be individual models and not comparable to prior studies if so. It is also unclear to me why those were chosen to be highlighted. In general, the Introduction could perhaps have more information about studies of attitudes and real examples of impacts on conservation efforts. Perhaps even a brief mention of there having been other studies of attitudes towards other animals that included bats as well may be useful, or some more discussion of conservation psychology. Or some information about the situation of bats in Chile—there really is very little in the introduction to provide some context for people and bats in the place of study so the urgency of this work in Chile is not clear.

Methods - It sounds rather odd to me that this is being called an “instrument” since that is vague—it may be better to say “assessment tool” or “survey” to be clearer about what deliverable the authors are presenting.

The Methods and Results section have verb tense inconsistencies in some paragraphs (switching between present and past tense within the same sentence) and needs to be re-edited.

Line 139 – “judgment of experts” – The two questions listed are not the only reason one should use experts to review the methods used to interview potential human subjects, but should be used to review methodology as a whole to ensure honesty and objectiveness. It is necessary to also ensure that questions or interview process were not asked in a leading manner or how the questions were truthfully answered. These particular matters are not addressed in the Methods section regarding the creation of this questionnaire and the interview process. I think having an expert on anthropology who has experience with interviewing human subjects should have been included on the panel, as this matter is rather standard in anthropology research methods.

  • The entire process between the pilot study leading to the finalized items that were used is not explained in a clear manner.
    • Line 155 – There needs to be a clearer explanation of what is meant by the “factorial structure did not agree exactly with the theoretical structure” before listing how the 5 types fared, and why those decisions were then reached for which were included.
    • Line 163 – Why were 39 items retained to the next stage despite the earlier section mentioning 35 items (and not all 51 if there was consideration of small sample size influencing the results)? There is no explanation here.

Line 166 – It is described that the participants were collected by convenience sampling, but it’s unclear to me how those participants were obtained (e.g. was it the university community like in the pilot study, were the participants residents in a certain area, details such as that would have been helpful to demonstrate that the population sampled were truly random). It may be better to move up the section about how the survey was conducted in order to better describe the participants who contributed.

Line 185-186 – Is there evidence that this was disseminated to a substantially large audience and not just snowball sampling from the initial posters (or of a few influential users)?

Results

There is a lot of repetition in how a section would refer to a table, repeat the numerical values, and then also place the entirety of the table in the main text. It makes it rather difficult to read the Results

Line 257 – I don’t quite understand what is meant by “The five items mentioned presented percentages between 75.1% and 86.7% in one response option

Line 262 – I’m not sure what item 19 is in reference to—there is not a single item in any table that is item 19.

Line 265 – Is there a significance to the 0.38 mentioned here as a threshold? It is not mentioned why this is important in any specific way or just a general statement about the data.

Spelling/Grammar

The references need to be re-read and corrected more thoroughly, as there are simple mistakes such as not putting the species name in italics and having the species in the taxonomic binominals be lowercase. There are also some citations that are just incomplete (e.g. #38).  

Line 56 – There’s a random “Rego” that ends the line after the citation.

Line 66 – I think it would be more accurate to say “anthropogenic”

Line 69 – Be consistent in how COVID-19 is capitalized or not, it was earlier written as “COVID-19” instead of “Covid-19” in the same paragraph.

Line 135-136 – That sentence is a bit awkward, might be better to rephrase it as “We selected and phrased questions using the most appropriate vocabulary within the Chilean cultural context.”

Line 152-154 – Those two sentences could be written to read more smoothly as “Subsequently, they provided researchers with feedback to determine which items needed modification or removal. Items removed were those that had low discriminative capacity or skewed data too heavily, resulting in the retention of 35 out of 51 items.” (re: where I suggest “skewed data” - I’m assuming that is what was meant by “contribution to the consistency of the scale” but am not sure since that is a rather odd turn of phrase.)

Line 166-170 – This paragraph should not be bolded.

Line 51 – “software” is a singular noun, not plural

Line 254 – “None of the items has a normal distribution” should be changed to “None of the items have a normal distribution”

Line 258 – “indicative of a low but not nil discriminative power”

Figures and Tables –

A repeated issue with the tables and figures is that there are captions that are not fully informative of what is being presented in them.

Table 1 should separate the different sociodemographic characteristics instead of listing everything as if it were a singular list. It is difficult to read since there is no variation not only in the font but even in the alignment of the section head labels.

Figure 1 – The caption about the factorial model is so overly simple that if I were to read it without reading the paper again, it would not make sense. A figure caption should provide at least a little more explanation and context so that if someone were to only see the figure and its caption, they should be able to understand it. Also, given the length of this paper, there are so many tables and figures—the listing of all these models seems extraneous and like it should be in Supplementary Material instead.

Table 2 – This table seems extraneous for the main text since it’s the totality of the descriptive data—it should probably be in Supplementary Materials.

Table 3 – This table shows the items that were included in the study, but I would have liked to see a full list of all the 53 items (mentioned in Line 140) in a supplementary document. Knowing what type of questions were not discriminative enough could be valuable to other researchers. I’m also unclear why some of the columns of values within each separate section for factors are bolded, which should have been described in the caption. The caption also says that sample group 1 is N=819, which is in disagreement with what is written in the Data Analysis section and every other mention of group one (N=820).

Table 5 – “All correlations are significant p<.000“ – This is an error from SPSS or other software—usually it means you need to double click to see the exact number (e.g. it may be 0.000001). If the journal standards are to report p values up to 3 decimals though, then this could be changed to p<0.001

(This error is also in various other places where p values were reported.)

Table 7 – the decimal points change to a comma instead part way through the table. I also think that the most important results of this table should be stated in the text but this complete table should be in a supplement.

Reviewer 2 Report

The study "Design and psychometric properties of BAtSS: A new
tool to assess attitudes towards bats" by Pérez et al. is an interesting research about the socio demographic factors related with attitudes towards bats. The manuscript is easy reading. However, the method section is difficult to follow. Because the statistical procedures are not matched with a prediction. In addition, if the goal is made their results comparative with other studies the statistical procedures will be open as a R o Phyton code.  

I have only one specific commment

Line 193-95. I did not understand the logic of this procedure. Could yo explain why yo did?

Reviewer 3 Report

Review of “Design and psychometric properties of BAtSS: A new tool to assess attitudes toward bats” for Animals

There are a few goals of this article; 1) to design a questionnaire with validity and internal consistency; 2) to ask Chilean participants their views on bats; and 3) to advocate for using this type of questionnaire as a way to change peoples’ attitudes toward bats (i.e., to enhance conservation efforts).  The article presents very detailed information about the construction of the tool, which may be of relatively less importance to readers of Animals than the information about peoples’ attitudes toward bats.  Although the authors do talk a little about the third goal, this part could be more fully developed as it is not clear how this questionnaire will be used to “solve human-wildlife conflicts” (line 30-31).

I could recommend that these three goals be more clearly demarcated in the text and that the different aspects of the first goal are more clearly differentiated.  In the development of the questionnaire, it would help the reader to be told 1) how pre-existing questionnaires were used for the current study (section 1.1 needs to include more explicit statements about relevance to the current study); 2) what was the procedure for deciding which items to retain versus which to discard; 3) why split the sample (section 2.4); and 4) why each statistic was being used and what the authors conclude as a result.  It currently is very statistical, and although I feel very confident in most statistical matters, I was completed lost in the current descriptions of the different tests, and what they told us about validity and internal consistency.  The reader should be told what conclusion follows from each of the tests used in the construction of the questionnaire.  I think that these are relatively simple, yet very important fixes.

Be careful not to use the acronym prior to defining it (e.g., EFA, CFA but there are more).

I really liked Figure 1 showing the different possible models.  I did not see the statistical comparison that revealed the ‘best’ model was M3.  (for example, lines 291 discusses models but is this the analysis related to which of the models shown in Fig 1 is the best?).  I am not sure why this model is presented again separately as Figure 2 (note that the explanations in Fig 2 should also be in Figure 1).  I also wasn’t sure the consequences /benefits /drawbacks of having the three negativity factors combine into one super negative category.  For example, does this mean that future users should consider only the singular negative category?

What does it mean that the correlation between the factors is significant, even though the factor analyses shows easily discriminable factors? 

The results of the design part should be distinct from the results of the second and third goals.  It is confusing to have all three parts intermingled as in the current ms (this is not such a simple fix but is also very important).

I recommend a little more time is spent on describing Chileans' attitudes toward bats in your large sample.

There is repeated mention of items that can be, or are difficult to modify.  While reading this in the results section, I interpreted it to mean that the wording of the item was important, or that people found it difficult to answer the question.  However, in the discussion it seemed that this referred to how easy it is to change people’s attitudes.  How was this determined?  I also had no idea what the parameters presented in Table 7 referred to.

Along with more time spent on saying what this study tells us about attitudes to bats, I think the results presented in Table 8 and Table 9 would be better presented as figures.

The discussion brings the results to a higher level, relating attitudes about bats to conservation efforts.  However, more information is needed.  I think it might also be important to link this study to more information about Chileans’ conservation behavior in general, as all we are told is their extreme aggression toward bats

Lines 375-377: specify which items or which factor of the questionnaire related specifically to aggression.  Explain how this relation links to support for conservation efforts.

Lines 378-380: explain how items that are more difficult to change relates to immediate danger for humans.

Line 386-389: I did not understand the point the authors are trying to make here.

Line 405:  Please explain both here and in the results section.

Overall, I like the paper. It appears to be very rigorous in the design of the questionnaire, but readers of the journal Animals will also want to know what this tells us about people’s attitudes toward bats.  Whether this study can tell us anything about how attitudes are related to behaviors can be discussed, and perhaps can be framed as more speculative (if I am are reading the discussion correctly).
